# The Use of Crystal Violet Degradation Products for Ballpoint Pen Ink Manuscript Dating

**DOI:** 10.3390/molecules28176429

**Published:** 2023-09-04

**Authors:** Óscar Díaz-Santana, Nuria Cárdenes-Sánchez, Francisco Conde-Hardisson, Argimiro Rivero-Rosales, Miguel Suárez de Tangil Navarro, Daura Vega-Moreno

**Affiliations:** 1Laboratorio de Grafística y Documentoscopia del Servicio de Criminalística, Instituto Canario de Análisis Criminológico (ICAC), 35001 Las Palmas, Spain; administracion@icac-canarias.com (Ó.D.-S.);; 2Agilent Technologies, 28232 Madrid, Spain; 3Departamento de Química, Universidad de Las Palmas de Gran Canaria (ULPGC), 35017 Las Palmas, Spain

**Keywords:** ballpoint pen ink, dyes, chromatography, manuscript dating, degradation, crystal violet

## Abstract

Determining the approximate dates that written documents were drawn up based on the chemical composition of the ink is not a simple process. It is very demanding in terms of legal requirements. Various studies have succeeded in dating manuscripts by analyzing the temporal evolutions of the concentrations of dyes and solvents in documents based on the original formulations of the ink pens. These analyses were carried out simultaneously by HPLC-DAD for dyes and by GC-MS for solvents. This study aims, for the first time, to evaluate novel ink compounds and the temporal evolution of the concentrations of the degradation products of the dyes used by most suppliers and which are present in almost all types of ballpoint inks, i.e., Crystal Violet (CV). CV degrades through two parallel pathways: on the one hand, it undergoes progressive demethylation until it becomes pararosaniline, and on the other, it undergoes a breakdown of the molecule obtaining, among other by-products, the compound *N,N′-Dimethyl-4-aminophenol* (NNAPH), that was experimentally verified using four different inks (e.g., Inoxcrom^®^ and Sigma^®^ brands, in blue and black). For the NNAPH compound, we observed that four of the inks under analysis displayed the same temporary behavior despite having different initial chemical compositions. These initial results show the high potential for both CV and NNAPH, together with the rest of the pararosaniline family, as age tracers for dated/old documents. These techniques may potentially open up new avenues for universal dating tools, regardless of the brands of ink employed for use in different ballpoint pen types.

## 1. Introduction

Ballpoint pen inks are composed mainly of solvents (~50%), dyes (~25%), resins, and additives (~25%) [1,2]. A standard pen can contain in its original composition a mixture of various types of solvents and dyes [3,4,5], which will vary in concentration once the ink is deposited onto paper [6]. For decades, there has been significant interest in the study of the temporal variability of the concentrations of these solvents and dyes [2,7,8,9]. This enables us to roughly date manuscripts and documents based on the relative concentrations of these compounds [10].

Using only solvents for this purpose, the temporal range in which this technique can be useful varies from a few months as the most frequent value [4] to a few years in the best case [11,12]. For this reason, there are dating methodologies that are based on combined studies of solvents and dyes and their evolution over time [5,13]. This allows an expansion of the time range over which the method can be applied [14]. These methodologies are focused on variations of the composition of ballpoint pen ink from compounds present in the initial compositions. Solvents decrease their concentration rapidly by evaporation and then stabilizing after a few months (maximum 2 years) [15,16]. The dye concentrations decrease more slowly, mainly due to degradation processes [17,18]. Some of them, like crystal violet (CV), maintain their concentrations to be relatively constant over time [19].

In some studies, there have been indications that by applying these types of dating methodologies, it is necessary to know the initial compositions of the ink that is to be dated; however, in judicial documents, this is a fact that is frequently unknown [6]. The initial compositions vary significantly from one pen color to another, but they also vary between different commercial brands of the same color. Color identifications are very easy to carry out with visual methods, but pen brand identification is not. To resolve the absence of trademark information, two mechanisms can be used: First is to evaluate the relative proportions between the different compounds and/or the existence of specific compounds used by a particular manufacturer that allows identification [20]. The other mechanism is to study the temporal evolution of the document under analysis in order to determine the age curves of the compounds that we are trying to date; this can be a lengthy procedure [11].

The degradation curves of the dyes and the evaporation of the solvents over time will vary depending on the environmental factors in which the document has been stored, such as light, temperature, or humidity [21]. In addition, because each type of ink has a significantly different initial composition, the analytical study of aging (through the variation in the concentration of the compounds with time) is highly complex, making it difficult to interpret the experimental results for a particular type of ink and, even more so, making it even more difficult to extrapolate results from other different inks [22]. It would be very useful to have more universal techniques or applications for the different types of inks in order to date these documents more easily [21].

CV is a widely used colorant that is used in 99.9% of ballpoint pen ink formulations for any color (black and blue) [11], which makes it a practically universal colorant for any commercial brand. The concentrations used in these inks are so high that the evolution of their concentrations over time is not readily appreciable; they are considered to have constant concentrations [23]. Frequently, the original ink compositions are not only CV but also two of its demethylated products: pentamethylpararosaniline (Penta-PRS) and, on many occasions, tetramethylpararosaniline (Tetra-PRS), collectively called Gentian Violet. If the existing concentrations of these three ink compounds are significantly high, it indicates that they were present in the original ink composition, and did not come only from the decomposition of the CV. However, in the remaining cases, these and other degradation products from the CV will have a much lower concentrations, which will vary with time, and may be useful as tracer compounds for the purpose of manuscript aging.

The CV of ballpoint inks has two main routes of decomposition over time once it has been deposited on paper. CV demethylation, in which the degradation products that appear are characterized by successive loss of methyl groups, from hexamethylpararosaniline (CV, with 6-CH_3_ groups) leading to pararosaniline (PRS) as the final product (any CH_3_ groups) [24]. These authors do not clearly specify the CV demethylation mechanism for this type of sample. One of the degradation mechanisms mentioned in the bibliography shows the liquid chromatography the presence of 8 intermediate compounds (in addition to the starting compound CV and the final compound PRS), but it would be a process of biodegradation by microorganisms [25]. To date, there are no analytical data that details and justifies the CV demethylation mechanism from ballpoint ink pens sampled on paper.

Another parallel reaction is the breakdown of CV into two compounds, Michler’s Ketone and *N,N′-dimethyl-4-aminophenol* (NNAPH) (Figure 1) [24]. Some authors suggest the formation of phenol as a degradation product of CV instead of the formation of NNAPH [26,27].

Both these processes can be forced or accelerated in cases of artificial aging of handwritten documents, frequently by subjecting them to heat sources or direct exposure to sunlight [1,2,28,29].

## 2. Results and Discussion

### 2.1. CV Demethylation Reaction

CV demethylation represents the successive loss of CH_3_ groups from CV (PRS with 6 CH_3_) to PRS (without any CH_3_) [24,30]. However, it is not known precisely how this demethylation process occurs in the intermediate compounds when CV degradation occurs on handwritten documents. It is known that demethylation arises by biological processes; a total of eight intermediate compounds are obtained [31], but the non-biological degradation of the molecule on paper, based on experimental data analyzed on four different inks (two blue and two black) by HPLC -DAD, show a total of five intermediate compounds in the demethylation of CV to PRS, previously described by Weyerman et al. [19] and experimentally demonstrated in this study. This differs from the demethylation produced by microorganisms.

The experimental data also indicate that the loss of the first three methyl groups (from Hexa-PRS to Tri-PRS) is significantly faster than the loss of the last three (from Tri-PRS to PRS) (see Table 1). The Tetra-PRS compound is present almost immediately from the deposition of the ink onto the paper, with a proportion of 4% with respect to the total pararosanilines. Both CV and Penta-PRS are present in the original composition of the ink, with percentages of relative proportions with respect to a total of 7 and 87%, respectively, at time 0. Over time, the CV maintains its constant concentration, but nevertheless, the relative percentage increases with respect to the total of the pararosanilines (+0.51%) for all the inks analyzed due to the decrease in total concentration (see Table 1 for the average values for the four inks that were analyzed).

In contrast, the Tri-PRS compound takes an average of 3.25 months to appear. However, less methylated compounds such as Mono-PRS and PRS have slower generation times of 7.25 months (for Mono-PRS) and greater than 96 months for PRS (corroborated with documents of recorded dates). The presence of Di-PRS from the initial deposition of the ink could suggest that there are other sources of generation in addition to the one described here.

These data could be justified with a reaction scheme with the first demethylation of each amino group occurring before the second, where demethylation of up to the Tri-PRS is more efficient than the demethylation from it (each of the amino groups will lose one methyl group before losing the second methyl group, see Figure 2). Furthermore, according to the experiments (see Table 1), demethylation up to Tri-PRS occurs more rapidly than subsequent demethylation reactions.

Starting with trimethyl–pararosaniline (Tri-PRS), the rate of loss of methyl groups is reduced. Following this mechanism, Tri-PRS has a methyl group in each of its amino groups, reducing the possibility of isomer formation with three other methyl groups, as occurs in cases of degradation by microorganisms.

As the ink on a document ages, the concentrations of Di-PRS increase, and Mono-PRS also appears. PRS compounds (without any CH_3_) only appear in cases where very old inks are being analyzed (>96 months, experimentally tested). Based on the experimental results, it can be observed that the presence of a high concentration of the demethylated compound or with few methyl groups indicates that it is an old ink (>96 months old).

In chromatograms obtained by liquid chromatography with UV-vis detector (HPLC-DAD at λ = 590 nm) for time = 0 (immediately after ink deposition) and for time = 96 months (Figure 1 for Blue Inoxcrom^©^ as an example), we can observe the different characteristic peaks of CV demethylation (seven peaks in total, although PRS at very low concentrations < LOQ). In more recent manuscripts, the compounds with fewer methyl groups are not significant; however, in older documents, the seven compounds can be observed, from CV to PRS (CV → Penta-PRS → Tetra-PRS → Tri-PRS → Di-PRS → Mono-PRS → PRS). If the demethylation reactions for these types of samples were combined, allowing other types of isomers as identified by Chen et al. (2007) [31], then a total of 10 peaks may appear in the chromatogram instead of the seven that appeared. But, these obtained experimental data demonstrate the type of reaction suggested.

### 2.2. CV Breakdown Reaction

In addition to the demethylation process described above, for CV, and also for the rest of the pararosaniline series, especially for Penta-PRS and Tetra-PRS, a second process occurs where the compound decomposes into two by-products.

In the breakdown of all these compounds (CV, Penta-PRS, and Tetra-PRS), the same subproduct is always produced: the same molecule, *N,N′-dimethyl-4-aminophenol* (NNAPH) with a molecular formula of C_8_H_11_NO (see Figure 3). This reaction involves oxygen from the air and moisture (H_2_O). The contribution of water molecules from the humidity of the air breaks the original molecule into two reaction products.

The other resulting sub-products will vary depending on the total number of methyl groups present in the starting compound. In the case of the CV decomposition, Michler’s Ketone is produced, but for Penta-PRS and Tetra-PRS, the produced compounds will be different (Methanone-C_16_ and Methanone-C_15_ compounds, respectively).

The generation of these compounds, including the PRS family, Michler’s Ketone, and NNAPH, has been experimentally determined for 27 different time points between time = 0 (ink freshly deposited on paper) and time = 96 months after deposition.

The NNAPH compound, a degradation product of the CV series, serves as a tracer for aging a handwritten document. The analysis of this compound in a questioned document, along with the determination of the rest of the series compounds and different CV degradation products, will be used to date the document. Additionally, since CV is a widely used dye, it can be applied as a practically universal dating method for ballpoint pen ink.

#### 2.2.1. NNAPH Temporal Evolution

From experimental data, a temporal cutoff line can be traced and analyzed from the evolution of the NNAPH concentration over time. In this evolution, a change in the relative concentration of NNAPH with respect to the CV concentration occurs between 49 and 60 months, depending on the type of pen. The concentration of the compounds must be normalized to a reference compound that does not significantly modify its concentration over time, such as CV. This is because its concentration is practically constant over time, and its degradation losses are considered insignificant compared to its total concentration. This ratio ([NNAPH]/[CV]) allows for analyzing the values independently of the sampled ink mass.

The concentration of NNAPH and CV was analyzed from time zero to 96 months, with a total of 27 samples for each ink type. Figure 2 shows the relative concentration of NNAPH over time obtained for the four analyzed inks (two blue inks and two black inks, Inoxcrom^®^ and Sigma^®^ brands). The observed cutoff line is between 49 and 60 months, where NNAPH remains at rest and in low concentrations.

To better visualize this effect, the same data can also be plotted inversely, representing the concentration of CV with respect to NNAPH (the values have been divided by 100 for better representation and quantification) (see Figure 3). Based on the experimental results, it is observed that if the analyzed ink is more than 49 months old, then the quantitative relationship between [CV]/(100 × NNAPH]) will be above 1. Conversely, values below 1 represent more recent inks.

For [CV]/(100 × [NNAPH]) values very close to 1, the results may not be conclusive. For the Sigma^®^ black ink, the cutoff point will not be the same value but a higher one, around 1.5. Therefore, to be used as a universal method for any type of ink brand and color, it will be necessary to expand the studies to more pen brands and make detailed distinctions for each color type.

#### 2.2.2. Correlations between the Different Degradation Compounds of CV

The analyzed experimental data show that at time zero, only the peaks of gentian violet (CV → Penta-PRS → Tetra-PRS) and a residual amount of Di-PRS exist in the inks. However, in 96-month-old inks, we already observe a variety of compounds that are degradation products of these original compounds (see Figure 1), such as Tri-PRS, Di-PRS, Mono-PRS, PRS, but also Michler’s Ketone and NNAPH, among others. The different degradation products will increase in relative concentration over time, as shown in Table 1.

Based on the obtained relative concentrations (normalized with the concentration of CV) for each of the compounds analyzed over 96 months (for 27 temporal points), the correlation study has been carried out between all the studied compounds, both those present in the original composition of the ink and those coming from the degradation of these (see Figure 4). For almost all analyzed inks, the distribution is non-parametric, except for the Sigma^®^ Blue ink, which follows a parametric correlation.

Appendix A represent the correlation values (with their corresponding *p*-value) for parametric (Pearson) and non-parametric (Tau B and Rho Spearman) correlations for all compounds of the four analyzed inks (see Appendix A). These experimental data show a different trend for each ink color, as the original composition will vary with color, but also between the brands used. The highest correlations between degradation compounds and the original ink compounds are observed in the Sigma^®^ brand, and within this, the correlation is higher for the black ink. On the other hand, the black Inoxcrom^®^ does not seem to have a high correlation between the different degradation products and the original ink compounds, possibly due to the very low concentrations presented by NNAPH for this ink from 56 months (see Figure 3), which determines a sudden change in the trend of this compound over time, and yet this abrupt change is not appreciated in the series of pararosanilines.

Regarding the validation of the reactions described for the degradation of CV, based on these experimental data, the following should be highlighted: (i) For blue inks, the correlation between Michler’s Ketone and NNAPH is highly significant, and in the case of Blue Sigma^®^, these two compounds are also correlated with the PRS family from which they may come (Penta-PRS and Tetra-PRS); these data could represent that this rupture reaction is the main one for this type of inks. (ii) On the contrary, black inks seem to have greater relevance for the demethylation process since the correlation between Penta-PRS, Tetra-PRS, Tri-PRS, and Di-PRS (and even Mono-PRS for Black Sigma^®^) is highly significant or significant in most cases.

Based on these results, it would be necessary to expand the studies to other ink brands so that with these additional data, predictive models can be constructed that allow the evaluation of the Time variable through multiple linear regression based on the concentration data of the different compounds, not only those present in the original ink composition (CV, Penta-PRS, and in some cases Tetra-PRS), but also based on the degradation compounds produced by the different reactions, such as Tetra-PRS, Tri-PRS, Di-PRS, Mono-PRS, PRS, as well as Michler’s Ketone, NNAPH, and the compound derived from the oxidation of the latter, NNDAB.

## 3. Material and Methods

### 3.1. Reagents and Standards

Methanol (HPLC gradient grade, 99.9%), acetonitrile (HPLC gradient grade, 99.8%), and HPLC-grade water were obtained from Sigma-Aldrich^®^ (St. Louis, MO, USA). To determine the solvents, dyes, and CV degradation products, a methanolic solution of 3-methylphenol (99.9%, Supelco^®^) was prepared as an internal standard (IS) at a concentration of 1000 µg·mL^−1^.

Analytical standards used for the calibration of all measured solvents were obtained from Sigma-Aldrich^®^ and Dr. Ehrenstorfer Laboratories^®^ (see Table 2).

However, for some compounds, such as Tri-PRS, Di-PRS, and PRS, there were no available standards for calibration. In these cases, the retention time was determined experimentally using ink samples, and the concentrations were analyzed using the Tetra-PRS standard as a reference value.

### 3.2. Ballpoint Pens Analyzed

Four ballpoint pens from two different companies, Inoxcrom^®^ and Sigma^®^, were analyzed: two with black ink and two with blue ink. The blue Inoxcrom^®^ pen had Fine Bille Style No. I.R.I. 08-145.682, Sierra IBB Fine blue ink (Spain—E0-028), and the black Inoxcrom^®^ pen had Fine Bille Style No. I.R.I. 08-145.682, Sierra IBB Fine black ink (Spain—E4-157). The blue Sigma^®^ pen had Ballpoint 1 mm, PX-6MA blue ink (Spain), and the black Sigma^®^ pen had Ballpoint 1 mm, PX-6MR black ink (Spain). Ink samples from these ballpoint pens were deposited monthly on white paper from Hewlett-Packard^®^ (80 g/m^2^, A_4_ format) for eight years prior to testing. The samples were stored in the dark in closed filing cabinets in a conditioned room between 21–23 °C. Relative humidity during storage stayed within the interval of 52–88%, but humidity is not a determining factor in storage conditions [32].

### 3.3. Sample Extraction Procedure

A Harris micro-punch with a diameter of 1.25 mm (Ø) and a Harris cutting mat was purchased from Sigma-Aldrich (St. Louis, MO, USA) and used for sampling. Eight microdiscs, each with a diameter of 1.25 mm, were used per sample. The extraction procedure was performed in sealed amber vials with 0.1 mL conical transparent glass inserts for analysis. The eight microdiscs were extracted with 20 µL of a methanol solution containing 0.1 mg·L^−1^ of the internal standard (3-methylphenol). The extraction involved agitating the sample in a closed vial with the extraction solution for 30 s using an orbital shaker. To determine the respective concentrations of solvents and dyes, the resulting extract was analyzed using both gas chromatography coupled to mass spectrometry (GC-MS) and high-pressure liquid chromatography with a diode array detector (HPLC-DAD).

Blank discs were cut from paper without ink and extracted using the same procedure to check for any contamination in the paper that might influence the ink aging results. The complete extraction process was carried out every month for 96 months to provide sufficient temporal resolution over the study period.

### 3.4. GC-MS Analysis

NNAPH, *N,N′-dimethyl-4-aminobenzaldehyde* (NNDAB), and Phenol (PH) were analyzed using GC-MS with an Agilent 8860A/5977B (Agilent Technologies^®^, Palo Alto, CA, USA) equipped with an Agilent 7650ALS autosampler. VF-WAXms 30 m × 0.25 mm i.d × 0.25 µm film thickness capillary columns (both from Agilent Technologies^®^) were used. Data acquisition was performed using MSD MassHunter software v.10.1, and quantitative analysis was carried out using the NIST 2017 v.2.0 Mass Spectral Library.

GC-MS analysis was performed with the following conditions: injection volume of 1 µL, splitless injection at 230 °C, GC column temperature starting at 45 °C, 1-min isothermal, 8 °C·min^−1^ to 230 °C, held for 5 min. The carrier gas was Helium (99.999%) with a constant flow rate of 1 mL·min^−1^. The temperature of the transfer line was maintained at 230 °C, and ionization was performed with impact electrons having kinetic energy levels of 70 eV. The temperatures of the source and quadrupole were 230 °C and 150 °C, respectively. The MS analysis was carried out in SIM mode on the selected ions that are characteristic of each compound: quantitation and qualifier ions for PH, NNDAB, and NNAPH were, respectively, 94 (66, 65), 148 (149), and 136 (137, 121), with a dwell time of 30.

Analytical parameters for the compounds measured by GC-MS are detailed in Table 3. The detection limits were calculated as three times the standard deviation of the residuals divided by the slope for the standard with a concentration of mg·L^−1^ (10 replicates). The detection limits of the solvents studied varied between 0.35 and 0.53 mg·L^−1^, and the corresponding quantitation limits were between 1.16 and 1.78 mg·L^−1^.

### 3.5. HPLC-DAD Analysis

For the quantitative analysis of dyes and some degradation products, an Agilent Series 1260 Infinity HPLC system (Agilent Technologies, Palo Alto, CA, USA) was used, which included a quaternary pump, an online degasser, an automatic injector, a thermostatted column compartment, and a DAD detector. Separation was carried out using an Agilent Poroshell 120 column (100 mm × 4.6 mm × 2.7 µm) at 25 °C. The mobile phase was composed of a solution of ammonium formate buffer at pH 4.0 in water (A) (190 µL formic acid and 0.64 g ammonium formate in 0.5 L water) and acetonitrile (B), using a linear gradient program from 20% to 100% acetonitrile over 32 min at a flow rate of 0.8 mL/min. The sample injection volume was 2 µL. The detection wavelength was set per compound to a value between 420 nm and 635 nm, according to the analyte’s absorption maximum (slit 4 nm). The dyes were identified by comparing their UV-Vis spectra and retention times with those of standards. Analytical parameters (retention time, RSD, limit of detection, limit of quantification) for the compounds analyzed by HPLC-DAD are detailed in Table 4. Data processing was carried out using OpenLab CDS ChemStation C.01.04.

### 3.6. Statistical Analysis

A statistical analysis was carried out using IBM SPSS Statistics for Windows, Version 26.0 (Armonk, NY, USA). A study of the normality of the variables was carried out in order to select the correlations, those of both a parametric nature and those of a non-parametric nature. First, a Pearson correlation was applied [33], which is a measure of linear dependence between two quantitative random variables, independent of the scales of measurement of the variables. This diagnostic is used to measure and quantify the linear dependence between two measured variables following parametric correlations. Second, Spearman’s Rho and Kendall’s Tau were also applied to the same values, assuming a non-parametric distribution. Spearman’s Rho is a non-parametric measure of dependency where correlation between two variables is studied (bivariate non-parametric correlation) [34]. Kendall’s Tau B coefficient is used to measure the ordinal association between two variables [35]. The interpretation is similar to Pearson’s correlation, but in this case, it is applied to ordinal variables and falls within the framework of non-parametric statistical techniques.

In all inferential statistical tests, it is considered significant when the *p*-value ≤ 0.05 (confidence level 5%) and highly significant when *p*-value ≤ 0.01 (confidence level 1%). Therefore, if *p* > 0.05, the test is not significant.

## 4. Conclusions

This study shows for the first time the relevance of the use of the degradation products of CV as dating tools for questioned documents. This is in contrast to what has been used up to now, i.e., temporal monitoring of the dyes and solvents present in the initial formulations of the ink (original compounds, not degraded).

The analysis of concentrated CV and other derivatives, such as Gentian Violet and its degradation products, may have great potential for manuscript dating because it broadens the temporal range of applicability in comparison with the use of solvents (such as 2-Phenoxyethanol, PE). The temporal evolution of dyes is slower and longer than that of solvents. These dyes and their degradation products vary between months to years (even decades), and yet, most of the solvents stabilize within a few months, with a maximum applicability of about 2 years. In addition, CV is the most widely used dye on the market, being present in almost all ballpoint inks. Therefore, its use as a dating tool could have almost universal applicability for all questioned documents written with ballpoint pen ink.

The family of degradation compounds that can be used as tracers of the aging of a document are basically two: (i) the compounds resulting from successive demethylation of CV and Penta-PRS (Tetra-PRS, Tri-PRS, Di-PRS, Mono-PRS, and PRS), and (ii) the cleavage of the CV and Penta-PRS molecule to give *N,N′-dimethyl-4-aminophenol* (NNAPH) and Michler’s Ketone (in case of proceeding from the cleavage of the CV).

We have observed that all these indicated compounds generally have a high correlation with each other, and hence, it is necessary to expand our studies with more samples and more brands of ballpoint ink in order to be able to establish multiple linear regression models between these compounds and using time as the dependent variable.

On the other hand, we have observed that NNAPH behaves similarly on different inks (in brand and color) despite having a different starting chemical composition. The relative concentration of this NNAPH compound with respect to CV makes it possible to establish a cut-off line for the four analyzed inks that makes it possible to distinguish between newer and older ink depositions (cut-off line established between 49 and 60 months).

These results are focused on the CV temporal evolution and its associated and/or degraded products, but it would be advisable to combine these results with the temporal evolution of different solvents and the presence and variation of other dyes. This combination will allow for a better temporal approximation, as in the first months after ink de-position, solvents are the compounds that provide the most information due to their rapid evaporation; however, their applicability diminishes over time. In the temporal range of several years, the applicability of this new dating methodology has the greatest potential.

Moreover, it is important to establish whether other brands of ballpoint pens of the NNAPH compound behave in the same way as has been observed for these four inks, as well as evaluate whether different storage conditions or subsequent paper treatment after writing (artificial aging) can alter this observed pattern. This increase in the number of analyzed samples will also make it possible to assess whether the high correlation rates between the different degradation compounds are maintained. With these data, we also need to carry out further regression model analyses that allow documents to be dated, regardless of the pen brands that were used.

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
