# Peer review of "The Use of Crystal Violet Degradation Products for Ballpoint Pen Ink Manuscript Dating"

_molecules, 2023, doi:10.3390/molecules28176429_

Round 1

Reviewer 1 Report

Overall Impressions

 The manuscript presents an interesting approach for dating ballpoint pen ink using the degradation products of crystal violet (CV), a dye present in most ballpoint inks.

The key findings are:

Two degradation pathways were characterized - demethylation of CV and cleavage to form NNAPH.

The relative concentrations of CV degradation products change over time in a predictable manner.

The NNAPH/CV ratio shows a clear shift at 49-60 months, providing a potential dating tool.

The methodology is sound, leveraging HPLC and GC-MS to quantify the ink components. A good number of samples were analyzed over an 8-year period.

The results support the proposed degradation pathways of CV and demonstrate the utility of using the degradation products for dating purposes.

Major Comments

 The Introduction provides good background but could be condensed in places to highlight the key gaps being addressed by the current study.

Parts of the Results/Discussion could be streamlined for clarity and flow. For example, the detailed discussion on the demethylation pathway could focus on the key implications for dating.

The Conclusions nicely summarize the main findings. Some additional discussion on applications, limitations, and future directions could further enhance this section.

Minor Comments

 Abbreviate crystal violet as CV after defining it initially.

Italicize compound names (e.g. N,N’-dimethyl-4-aminophenol)

Carefully proofread for minor errors.

Overall this is a well-executed study demonstrating a novel approach to ballpoint ink dating using CV degradation. The results indicate good potential for practical applications. Addressing the comments above would further improve the quality of the manuscript.

Author Response

The abbreviation for Crystal Violet (CV) was defined at Line 50, and after this line, “Crystal Violet” has been modified to CV. (Lines 73, 101, 198 & 310).

The format for compound names has been modified to italic (Line 22, 98, 202, 358 & 443).

The sentence “Based on the experimental results, it can be observed that the presence of a high concentration of the demethylated compound (PRS) or with few methyl groups (Mono-PRS and Di-PRS) indicates that it is an old ink (> 96 months old).” has been included at Results and Discussion section, at Lines 165-167.

A new paragraph has been included in Conclusions, summarizing the application and limitations of the results of this work. “These results are focused on the CV temporal evolution and its associated and/or degraded products, but it would be advisable to combine these results with the temporal evolution of different solvents and the presence and variation of other dyes. This combination will allow for a better temporal approximation, as in the first months after ink de-position, solvents are the compounds that provide the most information due to their rapid evaporation; however, their applicability diminishes over time. In the temporal range of several years is when the applicability of this new dating methodology has the greatest potential.” (Lines 455-462).

Reviewer 2 Report

REVIEW REPORT:

The paper is an interesting study on the dating of ballpoint ink manuscripts and presents a good analytical strategy for the characterisation and evaluation of the state of preservation of these inks, which are challenging due to their analytical complexity.

The proposed combination of HPLC-DAD and GC-MS techniques provides a broader perspective for the detection of the degradation compounds of CV and also for the study of their degradation pathways, comparing this strategy with others in the literature.

The study is well presented and structured with a clear exposition and justification of the results, and the statistical analysis carried out is complete.

Overall, this is a very good study with high relevance to the forensic field, suitable for publication in Molecules, provided the following considerations are taken into account:

- The number of figures is appropriate and supports the explanation of the results obtained, however, Figure 3 cited in the text is missing (page 9, line 335 and page 10, line 373). Please include it.

- Table 5 is also missing, although is cited in the text (page 10, line 355).

- Regarding the literature cited on the topic, there are some more recent investigations that are not reported and should be included. The results and improvements of the proposed analytical approach should be compared with the other strategies and commented on. The references to be included are:

* L. Ortiz-Herrero et al. Microchemical Journal 140 (2018) 158-166

* I. San Roman et al. Analytica Chimica Acta 892 (2015) 105-114

- Although the main novelty of this work is the complete and comprehensive characterisation of the dyes of the inks and their degradation products, the authors propose the combination of HPLC-DAD for this purpose and GC-MS for the analysis of the solvents present in the inks. With regard to the results obtained using this latter technique, only the quantitative parameters of LOD, LOQ and RSD are given, but the variation in concentration of the tree solvents identified in the manuscripts at different stages of ageing is not presented and commented on. This information should be included.

- As the authors point out, these are very promising results for dating questioned documents written with ballpoint pen ink, but in this investigation only two brands of two-colour inks (blue and black ink) were studied, so further studies should analyse other ballpoint pen inks to see if the behaviour of the NNAPH compound is similar to that observed for these. In addition, the documents analysed were stored under certain environmental conditions (RH and temperature) and these parameters may influence the degradation pathways of the materials studied, and the paper used may also influence the behaviour of the ink in the extraction process. For this reason, further studies with documents stored under different temperature and relative humidity conditions and with different types of paper are needed to evaluate the real potential of this method. Some comments on this issue should be included. Regarding the temperature and relative humidity conditions under which the documents studied were stored, these are not indicated in the text and this information should be included.

- Regarding the extraction process of the ink from the paper, only one extraction was performed, in which the sample was agitated in a closed vial with the extraction solution for 30 seconds using an orbital shaker. Has this process been optimised? (Have different agitation times been tested?) Some comments on this point should be included.

Author Response

Figure 3 and Table 5 (now Table 2) were indeed included on the manuscript and visible to Reviewer 1 and the Editor in the previous version, but for some reason, they were not visible to Reviewer 2 & 3 (I am unaware of the reason.). The updated version is now included also in .pdf format as well to ensure proper visualization.

References: L. Ortiz-Herrero et al. Microchemical Journal 140 (2018) 158-166; and  I. San Roman et al. Analytica Chimica Acta 892 (2015) 105-114, have been included on the manuscript (Introduction) as [9] and [12] respectively.

At Section 3 it has been included the sentence “The samples were stored in the dark in closed filing cabinets in a conditioned room between 21-23 ºC. Relative humidity during storage stayed within the interval 52–88%; but humidity is not a determining factor in storage conditions [32].” (Lines 338-341), providing the information requested by the reviewer. Furthermore, the provided details of the paper used have been reviewed, including the brand name and paper type (Line 337).

The extraction procedure was optimized in previous studies (2017) and is documented in the Doctoral Thesis of the first author of this work - Díaz-Santana (2017)- http://repositorio.ucjc.edu/handle/20.500.12020/247?locale-attribute=en- , including subsequent re-extraction procedures to study the efficiency, which was concluded to be over 98% with methanol (lower with acetonitrile). However, since ink traces are never homogeneous, the extraction efficiency is not a highly determining parameter on this kind of studies due to the results' dependence on the ink mass. For that reason, a detailed optimization for this process has not been included in the article. Therefore, it is necessary to always evaluate the relative concentration of target compounds (normalizing the results with respect to CV concentration).

Based on results of the optimization, the orbital shaker is not essential, but it accelerates the extraction process. Extraction time could be reduced to shorter times, but it's kept at 30 seconds to ensure complete extraction.

The results presented in this work have focused solely on the degradation products of CV and the compounds present from the initial moment. But all the degradation processes occurring in the inks can become quite complex and occur simultaneously in various ways, as ink is a highly intricate combination of different products, not just CV. The temporal evolution of phenol or NNDAB has not been provided in the results because the findings are inconclusive (the latter readily degrades into other compounds). The authors of this study have strong suspicions that, for instance, Phenol mainly originates from Phenoxyethanol (PE) among other solvents, not only from CV. Presenting the data outside of this context could lead to misinterpretations of the results. Currently, we are working on elucidating other pathways for the formation of such compounds (among others), by combining solvent data, dyes, and CV degradation products. However, these objectives are beyond the scope of this paper.
Nevertheless, if Reviewer 2 deems it necessary, they could be provided as supplementary material.

At conclusion, it has been included the following paragraph: “Moreover, it is important to establish whether other brands of ballpoint pens of the NNAPH compound behave in the same way as has been observed for these four inks, as well as evaluating whether different storage conditions or subsequent paper treatment after writing (artifitial aging) can alter this observed pattern.” (Lines 463-466).

Reviewer 3 Report

The manuscript entitled “The use of crystal violet degradation products for ballpoint pen 2 ink manuscript dating” and singed by Óscar Díaz-Santana at al. presents very interesting results, with importance for the readers dealing with legal or forensic document analysis. The manuscript aims to present the chemistry and analytical methodology for the degradation products of crystal violet, an almost ubiquitous dye in common inks. The study is interesting and worth publication but not before some relevant issues are addressed. Some positive appreciations and critics are indicated below:

Title and abstract: Clearly stated and the context/background, aims and main findings of the study are well presented.

Introduction: The first paragraphs present the typical composition of ballpoint pen inks with accents for legal analysis and main of the information focus on crystal violet (CV) which is welcome. The authors choose to insert a subsection 1.1 (CV aging in a handwritten document) but in my opinion, only one such subsection title is not necessary. In Scheme 1, in the chemical products there are two atoms of oxygen which cannot be found in the reactants, probably they come from molecular air oxygen (such is in Scheme 3), but the authors should indicate that for a scientific rigour.

Material and Methods: comprises of six, well organized subsections. The indicated information is well presented, and the experiments could be reproduced by any laboratory using the indicated information. The units of measurements for temperature (ËšC) should be rechecked for clarity, it seems that there is a problem with its writing. The statistical analysis section is also well presented.

Results and Discussion: It contains two subsections, (1) discussions for CV demethylation reactions and for (2) other CV breakdown reactions. What is the detection wavelength in the chromatograms from Figure 1? Since the spectral maximum differences for each analyte. It would have been better if the two chromatograms are superimposed wit the same frame for x axis. The authors do not discuss the differences in the demethylation process for other types of ballpoint inks. They mainly discuss the Blue Inoxcrom© ink case. What happened in the other cases? For example, for the other degradations, the authors do discuss the case for the other inks which appears to be different indeed.

The quality of Figure 2 is poor, the authors could increase the resolution of this figure. A zoomed-in insert for the first 12 months would be welcome.

Table 5 and Figure 3 are missing!!!

Page 10, line 356, what does it mean “but rather follows a non-parametric correlation”? Is there an asymmetric distribution? The term “non-parametric” is suited for statistical tests.

Conclusions although a bit lengthy, they are well written and supported by the results.

Author Response

Figure 3 and Table 5 (now Table 2) were indeed included on the manuscript and visible to Reviewer 1 and the Editor in the previous version, but for some reason, they were not visible to Reviewer 2 & 3 (I am unaware of the reason.). The updated version is now included in .pdf format as well to ensure proper visualization.

In the introduction, the existence of subsection 1.1 has been removed and it has been integrated with the rest of the text.

At Scheme 1, the following sentence has been included: “in pre-sence of atmospheric O2” (Line 101).

Just like what happened with Figure 3 and Table 5, there might be a formatting issue when reviewers view the document, as there is no error in the Celsius degrees display in the original version. Additionally, a .pdf version will be provided to ensure proper visualization.

The data has sufficient resolution to add a zoom to Figure 2 for the first 12 months; however, the data in this initial period for this particular compound is not highly relevant. This compound is of interest for longer time periods. For shorter time studies, the investigation of solvents is more relevant, as reflected in various bibliographic references (both our own and those of other authors). Adding the Figure for these first 12 months is not deemed appropriate, but it is being attached for the reviewer's assessment.

Figure 1: Chromatograms - Wavelength detection was 590 nm for CV and Penta-PRS, and 555 nm for the rest of the compounds. Chromatograms are shown at 590 nm (this data has been included at manuscript, Line 187.)

A same image with the two chromatograms superimposed with the same frame for x axis does not provide clear visibility of the distinct peaks of the compounds, as several of them overlap in an identical manner, making it impossible to clearly discern each line. For this reason, the visualization presented in Figure 1 has been provided.

As detailed in the text (see Line 126 “Table 1 for the average values for the 4 inks that were analyzed”), the provided results correspond to the 4 analyzed inks, but the chromatogram of one of them (Blue Inoxcrom) was added for simplification, since the results for the remaining three inks are also equivalent.

To clarify, the following sentence has been added to the text: “(Figure 1 for Blue Inoxcrom© as an example),” (Line 188)

The sentence “but rather follows a non-parametric correlation” has been modified by “For almost all analyzed inks the distribution is non-parametric, except for the Sigma® Blue ink that follows a parametric correlation.” (Lines 271-272).

Round 2

Reviewer 3 Report

The quality of the revised manuscript has been improved and could be accept for publication.